# Deep Perm-Set Net: Learn to predict sets with unknown permutation and cardinality using deep neural networks

## Abstract

Many real-world problems, *e.g.* object detection, have outputs that are naturally expressed as sets of entities. This creates a challenge for traditional deep neural networks which naturally deal with structured outputs such as vectors, matrices or tensors. We present a novel approach for learning to predict sets with unknown permutation and cardinality using deep neural networks. Specifically, in our formulation we incorporate the permutation as unobservable variable and estimate its distribution during the learning process using alternating optimization. We demonstrate the validity of this new formulation on two relevant vision problems: object detection, for which our formulation outperforms state-of-the-art detectors such as Faster R-CNN and YOLO, and a complex CAPTCHA test, where we observe that, surprisingly, our set based network acquired the ability of mimicking arithmetics without any rules being coded.

## 1 Introduction

Deep structured networks such as deep convolutional (CNN) and recurrent (RNN) neural networks have enjoyed great success in many real-world problems, including scene classification (8), semantic segmentation (16), speech recognition (6), gaming (12; 13), and image captioning (7). However, the current configuration of these networks is restricted to accept and predict structured inputs and outputs such as vectors, matrices, and tensors[1]. If the problem's inputs or/and outputs cannot be modelled in this way, these learning approaches all fail to learn a proper model (26). However, many real-world problems are naturally described as unstructured data such as sets (21; 26). A set is a collection of elements which is *invariant under permutation* and the size of a set is *not fixed* in advance. Set learning using deep networks is an emerging field of study that has generated substantial interest very recently (20; 21; 23; 26).

Consider the task of object detection as an example. Given a structured input, *e.g.* an image as a tensor, the goal is to predict a set of orderless locations, *e.g.* bounding boxes, from an unknown and varying number of objects. Therefore, the output of this problem can be properly modelled as a set of entities. However, a deep learning network cannot be simply trained to learn a proper model in order to directly predict unfixed number of orderless locations. Existing approaches formulate this problem using as a pre-defined and fixed-sized grid (18; 17) or anchor boxes (19) representing all possible locations and scales of the objects. Then, each location and scale is scored independently to contain an object or not. The final output is generated heuristically by a discretization process such as non-maximum suppression (NMS), which is not part of the learning process. Therefore, their performance is hindered by this heuristic procedure. To this end, the current solutions can only deal with moderate object occlusion. We argue that object detection problem can be properly formulated as a set prediction problem, where a deep learning network is trained to output a set of locations without any heuristic.

This shortcoming concerns not only object detection but also all problems where a set of instances (as input or/and output) is involved, *e.g.* a set of topics or concepts in documents (3), segmentation of object instances (10) and a set of trajectories in multiple object tracking (2). In contrast to problems such as classification, where the order of categories or the labels can be fixed during the training

---

[1]Throughout the paper, we use the term *structured data* for vectors, matrices and generally tensors which are fixed in size and ordered, *i.e.* any permutation of its values changes its meaning. This is not to be confused with *structured learning* which is commonly used for graphs.

process and the output can be well represented by a fixed-sized vector, the instances are often unfixed in number and orderless. More precisely, it does not matter in which order the instances are labelled and these labels do not relate the instances to a specific visual category. To this end, training a deep network to predict instances seems to be non-trivial. We believe that set learning is a principled solution to all of these problems.

In this paper, we present a novel approach for learning to deal with sets using deep learning. More clearly, in the presented model, we assume that the input (the observation) is still structured, *e.g.* an image, but the annotated output is available as a set. Our approach is inspired by a recent work on set learning using deep neural networks (21). Although the work in (21) handles orderless outputs in the testing/inference step, it requires ordered outputs in the learning step. This is a significant limitation, because in many applications, such as object detection, the outputs are naturally not ordered. When that happens, the approach in (21) cannot learn a sensible model (see Appendix for the experiment). In this paper, we propose a complete set prediction formulation to address this limitation. This provides a potential to tackle many more problems compared to (21).

The main contribution of the paper is summarised as follows:

1. We propose a principled formulation for neural networks to deal with arbitrary sets with unknown permutation and cardinality as available annotated outputs. This makes a neural network, for the first time, able to truly handle orderless outputs at both training and test time.
2. Additionally, our formulation allows us to learn the distribution over the unobservable permutation variables, which can be used to identify the most likely orders of the set. In some applications, there may exist one or several dominant orders, which are unknown from the annotations.
3. For the first time, we reformulate object detection as a set prediction problem, where a deep network is learned end-to-end to generate the detection outputs with no heuristic involved. We outperform the state-of-the art object detectors, such as Faster R-CNN and YOLO v2, on both simulated and real data with high level of occlusions.
4. We also demonstrate the applicability of our framework algorithm for a complex CAPTCHA test which can be formulated as a set prediction problem.

## 2   RELATED WORK

Handling unstructured input and output data, such as sets or point patterns, for both learning and inference is an emerging field of study that has generated substantial interest in recent years. Approaches such as mixture models (3; 5; 22), learning distributions from a set of samples (14; 15), model-based multiple instance learning (25) and novelty detection from point pattern data (24), can be counted as few out many examples that use point patterns or sets as input or output and directly or indirectly model the set distributions. However, existing approaches often rely on parametric models, *e.g. i.i.d.* Poisson point or Gaussian Process assumptions (1; 24). Recently, deep learning has enabled us to use less parametric models to capture highly complex mapping distributions between structured inputs and outputs. Somewhat surprisingly, there are only few works on learning sets using deep neural networks. One interesting exception in this direction is the recent work of (23), which uses an RNN to read and predict sets. However, the output is still assumed to have a single order, which contradicts the orderless property of sets. Moreover, the framework can be used in combination with RNNs only and cannot be trivially extended to any arbitrary learning framework such as feed-forward architectures. Another recent work proposed by (26) is a deep learning framework which can deal with sets as input with different sizes and permutations. However, the outputs are either assumed to be structured, *e.g.* a scalar as a score, or a set with the same entities of the input set, which prevents this approach to be used for the problems that require output sets with arbitrary entities. Perhaps the most related work to our problem is a deep set network recently proposed by (21) which seamlessly integrates a deep learning framework into set learning in order to learn to output sets. However, the approach only formulates the outputs with unknown cardinality and does not consider the permutation variables of sets in the learning step. Therefore, its application is limited to the problems with a fixed order output such as image tagging and diverges when trying to learn unordered output sets as for the object detection problem. In this paper, we incorporate these permutations as unobservable variables in our formulation, and estimate their distribution during the learning process. This technical extension makes our proposed framework the only existing approach in literature which can truly learn to predict sets with arbitrary entities and permutations. It has the potential to reformulate some

of the existing problems, such as object detection, and to tackle a set of new applications, such as a logical CAPTCHA test which cannot be trivially solved by the existing architectures.

# 3 DEEP PERM-SET NETWORK

A set is a collection of elements which is invariant under permutation and the size of a set is not fixed in advance, *i.e.* $\mathcal{Y} = \{\mathbf{y}_1, \cdots, \mathbf{y}_m\}, m \in \mathbb{N}^*$. A statistical function describing a finite-set variable $\mathcal{Y}$ is a combinatorial probability density function $p(\mathcal{Y})$ defined by $p(\mathcal{Y}) = p(m)U^m p_m(\{\mathbf{y}_1, \mathbf{y}_2, \cdots, \mathbf{y}_m\})$, where $p(m)$ is the cardinality distribution of the set $\mathcal{Y}$ and $p_m(\{\mathbf{y}_1, \mathbf{y}_2, \cdots, \mathbf{y}_m\})$ is a symmetric joint probability density distribution of the set given known cardinality $m$. $U$ is the unit of hyper-volume in the feature space, which cancels out the unit of the probability density $p_m(\cdot)$ making it unit-less, and thereby avoids the unit mismatch across the different dimensions (cardinalities) (25).

Throughout the paper, we use $\mathcal{Y} = \{\mathbf{y}_1, \cdots, \mathbf{y}_m\}$ for a set with unknown cardinality and permutation, $\mathcal{Y}^m = \{\mathbf{y}_1, \cdots, \mathbf{y}_m\}^m$ for a set with known cardinality $m$ but unknown permutation and $\mathbf{Y}_{\boldsymbol{\pi}}^m = (\mathbf{y}_{\pi_1}, \cdots, \mathbf{y}_{\pi_m})$ for an ordered set with known cardinality (or dimension) $m$ and permutation $\boldsymbol{\pi}$, which means that the $m$ set elements are ordered under the permutation vector $\boldsymbol{\pi} = (\pi_1, \pi_2, \ldots, \pi_m)$. Note that an ordered set with known dimension and permutation is exactly a structured data such as vector, matrix and tensor.

According to the permutation invariant property of the sets, the set $\mathcal{Y}^m$ with known cardinality $m$ can be expressed by an ordered set with any arbitrary permutation, *i.e.* $\mathcal{Y}^m := \{\mathbf{Y}_{\boldsymbol{\pi}}^m | \forall \boldsymbol{\pi} \in \boldsymbol{\Pi}\}$, where, $\boldsymbol{\Pi}$ is the space of all feasible permutation $\boldsymbol{\Pi} = \{\boldsymbol{\pi}_1, \boldsymbol{\pi}_2, \cdots, \boldsymbol{\pi}_{m!}\}$ and $|\boldsymbol{\Pi}| := m!$. Therefore, the probability density of a set $\mathcal{Y}$ with unknown permutation and cardinality conditioned on the input $\mathbf{x}$ and the model parameters $\mathbf{w}$ is defined as

$$\begin{aligned} p(\mathcal{Y}|\mathbf{x}, \mathbf{w}) &= p(m|\mathbf{x}, \mathbf{w}) \times U^m \times p_m(\mathcal{Y}^m|\mathbf{x}, \mathbf{w}), \\ &= p(m|\mathbf{x}, \mathbf{w}) \times U^m \times \sum_{\forall \boldsymbol{\pi} \in \boldsymbol{\Pi}} p_m(\mathbf{Y}_{\boldsymbol{\pi}}^m, \boldsymbol{\pi}|\mathbf{x}, \mathbf{w}). \end{aligned} \tag{1}$$

The parameters $\mathbf{w}$ models both the *cardinality* distribution of the set elements $p(m|\cdot)$ as well as the joint state distribution of set elements and their *permutation* for a fixed cardinality $p_m(\mathbf{Y}_{\boldsymbol{\pi}}^m, \boldsymbol{\pi}|\cdot)$.

The above formulation represents the probability density of a set which is very general and completely independent of the choices of cardinality, state and permutation distributions. It is thus straightforward to transfer it to many applications that require the output to be a set. Definition of these distributions for the applications in this paper will be elaborated later.

## 3.1 POSTERIOR DISTRIBUTION

Given a training set $\mathcal{D} = \{(\mathbf{x}_i, \mathcal{Y}_i)\}$, where each training sample $i = 1, \ldots, n$ is a pair consisting of an input feature (*e.g.* image), $\mathbf{x}_i \in \mathbb{R}^l$ and an output set $\mathcal{Y}_i = \{\mathbf{y}_1, \mathbf{y}_2, \ldots, \mathbf{y}_{m_i}\}, \mathbf{y}_k \in \mathbb{R}^d, m_i \in \mathbb{N}^*$, the aim to learn the parameters $\mathbf{w}$ to estimate the set distribution in Eq. (1) using the training samples.

To learn the parameters $\mathbf{w}$, we assume that the training samples are independent from each other and the distribution $p(\mathbf{x})$ from which the input data is sampled is independent from both the output and the parameters. Then, the posterior distribution over the parameters can be derived as $p(\mathbf{w}|\mathcal{D}) \propto$

$$p(\mathcal{D}|\mathbf{w})p(\mathbf{w}) \propto \prod_{i=1}^{n} \left[ p(m_i|\mathbf{x}_i, \mathbf{w}) \times U^{m_i} \times \sum_{\forall \boldsymbol{\pi} \in \boldsymbol{\Pi}} p_m(\boldsymbol{\pi}|\mathbf{x}_i, \mathbf{w}) \times p_m(\mathbf{Y}_{\boldsymbol{\pi}}^{m_i}|\mathbf{x}_i, \mathbf{w}, \boldsymbol{\pi}) \right] p(\mathbf{w}).$$

Note that $p_m(\mathbf{Y}_{\boldsymbol{\pi}}^{m_i}, \boldsymbol{\pi}|\cdot)$ is decomposed according to the chain rule and $p(\mathbf{x})$ is eliminated as it appears in both the numerator and denominator. We also assume that the outputs in the set are derived from an independent and identically distributed (*i.i.d.*)-cluster point process model. Therefore, the full posterior distribution can be written as

$$p(\mathbf{w}|\mathcal{D}) \propto \prod_{i=1}^{n} \left[ p(m_i|\mathbf{x}_i, \mathbf{w}) \times U^{m_i} \times \sum_{\forall \boldsymbol{\pi} \in \boldsymbol{\Pi}} \left( p_m(\boldsymbol{\pi}|\mathbf{x}_i, \mathbf{w}) \times \prod_{\sigma=\pi_1}^{\pi_{m_i}} p(\mathbf{y}_\sigma|\mathbf{x}_i, \mathbf{w}, \boldsymbol{\pi}) \right) \right] p(\mathbf{w}). \tag{2}$$

In this paper, we use two categorical distributions to define cardinality $p(m_i|\cdot, \cdot)$ and permutation $p_m(\boldsymbol{\pi}|\cdot, \cdot)$ terms. However depending of the application, any discrete distribution such as Poisson,

binomial, negative binomial or Dirichlet-categorical (*cf.* (20; 21)), can be used for these terms. Moreover, we find the assumption about *i.i.d.* cluster point process practical for the reported applications. Nevertheless, the extension to non-*i.i.d.* cluster point process model for any other application would be a potential research direction for this work.

## 3.2 LEARNING

For learning the parameters, we use a point estimate for the posterior, *i.e.* $p(\mathbf{w}|\mathcal{D}) = \delta(\mathbf{w} = \mathbf{w}^*|\mathcal{D})$, where $\mathbf{w}^*$ is computed using the MAP estimator, *i.e.* $\mathbf{w}^* = \arg\min_{\mathbf{w}} -\log(p(\mathbf{w}|\mathcal{D}))$. Since $\mathbf{w}$ in this paper is assumed to be the parameters of a deep neural network, to estimate $\mathbf{w}^*$, we use commonly used stochastic gradient decent (SGD), $\mathbf{w}_k = \mathbf{w}_{k-1} - \eta \frac{-\partial \log(p(\mathbf{w}_{k-1}|\mathcal{D}))}{\partial \mathbf{w}_{k-1}}$, where $\eta$ is the learning rate. Moreover during learning procedure, we approximate the marginalization over all permutations, which can be infeasible for large permutation space, with the most significant permutations for each training instance, *i.e.*

$$p_m(\boldsymbol{\pi}|\mathbf{x}_i, \mathbf{w}) = \sum_{\forall \boldsymbol{\pi} \in \boldsymbol{\Pi}} \omega_{\boldsymbol{\pi}}(\mathbf{x}_i, \mathbf{w})\delta(\boldsymbol{\pi}) \approx \frac{1}{N_\kappa} \sum_{\forall \boldsymbol{\pi}_{i,k}^* \in \boldsymbol{\Pi}} \tilde{\omega}_{\boldsymbol{\pi}_{i,k}^*}(\mathbf{x}_i, \mathbf{w})\delta(\boldsymbol{\pi}_{i,k}^*), \tag{3}$$

where $\delta(\cdot)$ is Kronecker delta and $\sum_{\forall \boldsymbol{\pi} \in \boldsymbol{\Pi}} \omega_{\boldsymbol{\pi}}(\cdot, \cdot) = 1$. $\boldsymbol{\pi}_{i,k}^*$ is the most significant permutation for the training instance $i$, sampled from $p_m(\boldsymbol{\pi}|\cdot, \cdot) \times \prod_{\sigma=\pi_1}^{\pi_{m_i}} p(\mathbf{y}_\sigma|\cdot, \cdot, \boldsymbol{\pi})$ during $k^{th}$ iteration of SGD (using Eq. 5). The weight $\tilde{\omega}_{\boldsymbol{\pi}_{i,k}^*}(\cdot, \cdot)$ is proportional to the number of the same permutation samples $\boldsymbol{\pi}_{i,k}^*(\cdot, \cdot)$, extracted during all SGD iterations for the training instance $i$ and $N_\kappa$ is the total number of SGD iterations. Therefore, $\sum_{\forall \boldsymbol{\pi}_{i,k}^* \in \boldsymbol{\Pi}} \tilde{\omega}_{\boldsymbol{\pi}_{i,k}^*}(\cdot, \cdot)/N_\kappa = 1$. Note that at every iteration, as the parameter $\mathbf{w}$ updates, the best permutation $\boldsymbol{\pi}_{i,k}^*$ can change accordingly even for the same instance $\mathbf{x}_i$. This allows the network to traverse through the entire space $\boldsymbol{\Pi}$ and to approximate $p_m(\boldsymbol{\pi}|\mathbf{x}_i, \mathbf{w})$ by a set of significant permutations. To this end, $p_m(\boldsymbol{\pi}|\mathbf{x}_i, \mathbf{w})$ is assumed to be point estimates for each iteration of SGD. Therefore,

$$p(\mathbf{w}_k|\mathcal{D}) \;\propto\; \prod_{i=1}^{n} \left[ p(m_i|\mathbf{x}_i, \mathbf{w}_k) \times U^{m_i} \times \tilde{\omega}_{\boldsymbol{\pi}_{i,k}^*}(\mathbf{x}_i, \mathbf{w}_k) \times \prod_{\sigma=\pi_1}^{\pi_{m_i}} p(\mathbf{y}_\sigma|\mathbf{x}_i, \mathbf{w}_k, \boldsymbol{\pi}_{i,k}^*) \right] p(\mathbf{w}_k). \tag{4}$$

To compute $\mathbf{w}_k$ and $\boldsymbol{\pi}_{i,k}^*$, we use alternating optimization and use standard backpropagation to learn the parameters of the deep neural network.

$$\boldsymbol{\pi}_{i,k}^* = \arg\min_{\boldsymbol{\pi} \in \boldsymbol{\Pi}} \quad f_1\left(\mathbf{Y}_{\boldsymbol{\pi}}^{m_i}, \mathbf{O}_1(\mathbf{x}_i, \mathbf{w}_{k-1})\right) + f_2\left(\boldsymbol{\pi}, \mathbf{O}_2(\mathbf{x}_i, \mathbf{w}_{k-1})\right), \tag{5}$$

$$\mathbf{w}_k = \mathbf{w}_{k-1} - \eta \sum_{i=1}^{n} \left[ \frac{\partial f_1\left(\mathbf{Y}_{\boldsymbol{\pi}_{i,k}^*}^{m_i}, \mathbf{O}_1\right)}{\partial \mathbf{O}_1} \cdot \frac{\partial \mathbf{O}_1}{\partial \mathbf{w}} + \frac{\partial f_2\left(\boldsymbol{\pi}_{i,k}^*, \mathbf{O}_2\right)}{\partial \mathbf{O}_2} \cdot \frac{\partial \mathbf{O}_2}{\partial \mathbf{w}} + \frac{\partial f_3\left(m_i, \boldsymbol{\alpha}\right)}{\partial \boldsymbol{\alpha}} \cdot \frac{\partial \boldsymbol{\alpha}}{\partial \mathbf{w}} \right] + 2\gamma \mathbf{w} \tag{6}$$

where $\gamma$ is the regularization parameter, $f_1\left(\mathbf{Y}_{\boldsymbol{\pi}}^{m_i}, \mathbf{O}_1(\mathbf{x}_i, \mathbf{w})\right) = -\sum_{\sigma=\pi_1}^{\pi_{m_i}} \log\left(p(\mathbf{y}_\sigma|\mathbf{x}_i, \mathbf{w}, \boldsymbol{\pi})\right)$, $f_2\left(\boldsymbol{\pi}, \mathbf{O}_2(\mathbf{x}_i, \mathbf{w})\right) = -\log\left(\tilde{\omega}_{\boldsymbol{\pi}}(\mathbf{x}_i, \mathbf{w})\right)$, and $f_3\left(m_i, \boldsymbol{\alpha}(\mathbf{x}_i, \mathbf{w})\right) = -\log\left(p(m_i|\mathbf{x}_i, \mathbf{w})\right)$, where $\boldsymbol{\alpha}(\cdot, \cdot)$, $\mathbf{O}_1(\cdot, \cdot)$ and $\mathbf{O}_2(\cdot, \cdot)$ represent the part of output layer of the network, which respectively predict the cardinality, the states and the permutation of the set elements (Fig. 1).

Note that Eq. 5 is a discrete optimization to find the best permutation $\boldsymbol{\pi}_{i,k}^*$, *i.e.* the best unique assignment of ground truth to the output of the networks, which can be attained (Optimally or sub-optimally) using any independent discrete optimization approach. Therefore, the quality of its solution depends on the description of $f_1$ and $f_2$ and the solver. In this paper, since we assume that the set elements are *i.i.d.*, therefore $f_1$ would be a linear objective. Empirically, in our applications, we found out that estimation of the permutations from just $f_1$ is sufficient to train the network properly. In this case, the permutation can be optimally found in each iteration in polynomial time using the Hungarian algorithm. Finally, $\boldsymbol{\pi}_{i,k}^*$ representing a permutation sample, is used as the ground truth to update $f_2(\cdot)$ and to sort the elements of the ground truth set's state in the $f_1(\cdot)$ term in Eq. 6.

## 3.3 INFERENCE

Having learned the network parameters $\mathbf{w}^*$, for a test input $\mathbf{x}^+$, we use a MAP estimate to generate a set output, *i.e.* $\mathcal{Y}^* = \arg\min_{\mathcal{Y}} -\log\left(p(\mathcal{Y}|\mathcal{D}, \mathbf{x}^+, \mathbf{w}^*)\right)$

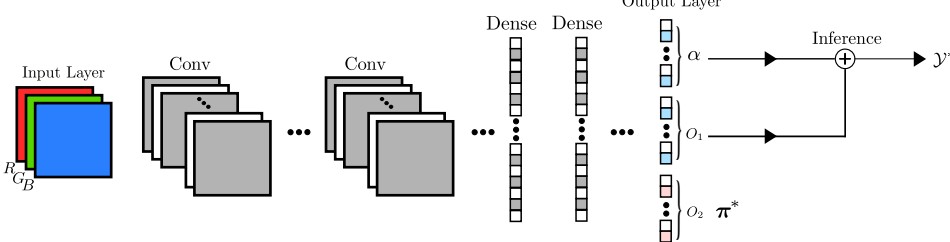

Figure 1: A schematic for our Deep Perm-Set Network. A structured input, *e.g.* an RGB image, is fed to a series of convolutional and fully connected layers with a collection of parameters shown by $\mathbf{w}$. The output layer consists of three parts shown by $\boldsymbol{\alpha}$, $\mathbf{O}_1$ and $\mathbf{O}_2$, which respectively predict the cardinality, the states and the permutation of the set elements. During the training, $\boldsymbol{\pi}_{i,k}^*$ representing a permutation sample (attained by Eq. 5), is used as the ground truth to update the loss $f_2(\boldsymbol{\pi}_{i,k}^*, \mathbf{O}_2)$ and to sort the elements of the ground truth set's state in the $f_1(\mathbf{Y}_{\boldsymbol{\pi}_{i,k}^*}, \mathbf{O}_1)$ term in Eq. 6. During inference, the optimal set $\mathcal{Y}^*$ is only calculated using the cardinality $\boldsymbol{\alpha}$ and the states $\mathbf{O}_1$ outputs. $\boldsymbol{\pi}^*$ is an extra output for ordering representation.

$$\mathcal{Y}^* = \underset{m, \mathcal{Y}^m}{\arg\min} \ -\log\left(p(m|\mathbf{x}^+, \mathbf{w}^*)\right) - m\log U - \log\sum_{\boldsymbol{\pi}\in\boldsymbol{\Pi}}\left(p_m(\boldsymbol{\pi}|\mathbf{x}^+, \mathbf{w}^*) \times \prod_{\sigma=\pi_1}^{\pi_m} p(\mathbf{y}_\sigma|\mathbf{x}^+, \mathbf{w}^*, \boldsymbol{\pi})\right).$$

Note that in contrast to the learning step, the way which the set elements during the prediction step are sorted and represented, will not affect the output values and therefore, the product $\prod_{\sigma=\pi_1}^{\pi_m} p(\mathbf{y}_\sigma|\mathbf{x}^+, \mathbf{w}^*, \boldsymbol{\pi})$ is exactly same for any permutation, *i.e.* $\forall \boldsymbol{\pi} \in \boldsymbol{\Pi}$. Therefore, it can be factorized from the summation, *i.e.* $\log\sum_{\boldsymbol{\pi}\in\boldsymbol{\Pi}}\left(p_m(\boldsymbol{\pi}|\mathbf{x}^+, \mathbf{w}^*) \times \prod_{\sigma=\pi_1}^{\pi_m} p(\mathbf{y}_\sigma|\mathbf{x}^+, \mathbf{w}^*, \boldsymbol{\pi})\right)$

$$= \log\left(\prod_{\sigma=1}^{m} p(\mathbf{y}_\sigma|\mathbf{x}^+, \mathbf{w}^*) \times \underbrace{\sum_{\boldsymbol{\pi}\in\boldsymbol{\Pi}} p_m(\boldsymbol{\pi}|\mathbf{x}^+, \mathbf{w}^*)}_{=1}\right) = \sum_{\sigma=1}^{m} \log\left(p(\mathbf{y}_\sigma|\mathbf{x}^+, \mathbf{w}^*)\right).$$

Therefore, the inference will be simplified to

$$\mathcal{Y}^* = \underset{m, \mathcal{Y}^m}{\arg\min} \ -\log\big(\underbrace{p(m|\mathbf{x}^+, \mathbf{w}^*)}_{\boldsymbol{\alpha}}\big) - m\log U \ - \sum_{\sigma=1}^{m} \log\big(\underbrace{p(\mathbf{y}_\sigma|\mathbf{x}^+, \mathbf{w}^*)}_{\mathbf{O}_1^\sigma}\big). \tag{7}$$

The above problem is the same inference problem as (21) and therefore can be optimally and efficiently calculated to find the most likely set $\mathcal{Y}^* = (m^*, \mathcal{Y}^{m^*})$. The unit of hyper-volume $U$ is assumed as a constant hyper-parameter, estimated from the validation set of the data.

As mentioned before, the distribution $p_m(\boldsymbol{\pi}|\cdot, \cdot)$ is approximated during the learning procedure by the samples $\boldsymbol{\pi}_{i,k}^*$ attained from Eq. (5). Depending on application, $p_m(\boldsymbol{\pi}|\cdot, \cdot)$ can be a single modal, multimodal or uniform distribution over the permutations. In any cases for ordering representation, the best permutation can be used, *i.e.* $\boldsymbol{\pi}^* = \arg\max_{\boldsymbol{\pi}\in\boldsymbol{\Pi}} \ p_m(\boldsymbol{\pi}|\mathbf{x}^+, \mathbf{w}^*)$.

A schematic for our set prediction neural network has been shown in Fig. 1.

## 4 EXPERIMENTAL RESULTS

To validate our proposed set learning approach, we perform experiments on two relevant applications including *i)* object detection and *ii)* a CAPTCHA test to perform de-summation operation for a digit from a set of digits. Both are appropriate applications for our model as their outputs are expected to be in the form of a set, either a set of locations or a set of candidate digits, with unknown cardinality and permutation.

### 4.1 OBJECT DETECTION.

Our first experiment is used to test our set formulation for the task of pedestrian detection. We compare it with the state-of-the art object detectors, *i.e.* Faster-RCNN (19) and YOLO v2 (17). To

ensure a fair comparison, we use the exactly same base network structure (ResNet-101) and train them on the same training dataset[2].

**Dataset.** We use training sequences from *MOTChallenge* pedestrian detection and tracking benchmark, *i.e.* 2DMOT2015 (9) and MOT17Det (11), to create our dataset. Each sequence is split into train and test sub-sequences, and the images are cropped on different scales so that each crop contains up to 4 pedestrians. Our aim is to show the main weakness of the existing object detectors, *i.e.* the heuristics used for handling partial occlusions, which is crucial in problems like multiple object tracking and instance level segmentation. To this end, we only evaluate the approaches on small-scale data which include a single type of objects, *e.g.* pedestrian, with high level of occlusions. The resulting dataset has 50K training and 5K test samples.

**Formulation.** We formulate the object detection as a set prediction problem $\mathcal{Y} = \{\mathbf{y}_1, \cdots, \mathbf{y}_m\}$, where each set element represents a bounding box as $\mathbf{y} = (x, y, w, h, s) \in \mathbb{R}^5$, where $(x, y)$ and $(w, h)$ are respectively the bounding boxes' position and size and $s$ represents an existence score for this set element.

**Training.** We train a convolutional neural network based on a standard ResNet-101 architecture, with loss heads directly attached to the the output of the ResNet. According to Eqs. (5, 6), there are three main terms (losses) that need to be defined for this task. Firstly, the state loss $f_1(\cdot)$ consists of two parts: *i)* Smooth $L_1$-loss for the bounding box regression between the predicted output states and the permuted ground truth states, *ii)* Binary cross-entropy loss for the presence scores $s$. The ground truth score for a specific instance is 1 if it exists and 0 otherwise. The permutation is estimated iteratively using alternation according to Eq. (5) using Hungarian (Munkres) algorithm. Secondly, a categorical loss (Softmax) is used for the permutation $f_2(\cdot)$. Since this variable is not observable from the annotations, $\pi^*_{i,k}$ is calculated to estimate the ground truth permutation(s). Finally, for cardinality $f_3(\cdot)$, a categorical loss is used in a similar fashion.

For training we use Adam optimizer with learning rate of 0.001, $\beta_1 = 0.9$, $\beta_2 = 0.999$ and $\epsilon = 10^{-8}$. To accelerate and stabilize the training process, batch normalization is employed and a weight decay of 0.0001 was used as an additional regularization term. The hyper-parameter $U$ is set to be 0.1, adjusted on the validation set.

**Evaluation protocol.** To quantify the detection performance, we adopt the commonly used evaluation curves and metrics (4) such as ROC, precision-recall curves, average precision (AP) and the log-average miss rate (MR) over false positive per image. Additionally, we compute the F1 score (the harmonic mean of precision and recall) for all competing methods.

**Detection results.** Quantitative detection results for Faster-RCNN, YOLO v2 and our proposed detector are shown in Tab. 1. Since our detector generates a single set only (a set of bounding boxes) using the inference introduced in Sec. 3.3, there exists one single value for precision-recall and thus F1-score. For this reason, the average precision (AP) and log-average miss rate (MR) calculated over different thresholds cannot be reported

Table 1: Detection results on the real data measured by average precision, the best F1 scores (higher is better) and log-average miss rate (lower is better).

| Method | AP ↑ | F1-score↑ | MR ↓ |
|---|---|---|---|
| Faster-RCNN | 0.68 | 0.76 | 0.48 |
| YOLO v2 | 0.68 | 0.76 | 0.48 |
| **Our Detector (w/o card.)** | **0.75** | **0.80** | **0.47** |
| **Our Detector** | – | **0.80** | |

in this case. To this end, we report these values on our approach using the predicted boxes with their scores only, and ignore the cardinality term and the inference step. To ensure a fair comparison, the F1-score reported in this table reflects the best score for Faster-RCNN and YOLO v2 along the precision-recall curve.

The quantitative results of Tab. 1 show that our detector using the set formulation significantly outperforms all other approaches on all metrics. We further investigate the failure cases for Faster-RCNN and YOLO v2 in Fig. 2. In case of heavy occlusions, the conventional formulation of both methods, which include the NMS heuristics, is not capable of correctly detecting all objects, *i.e.* pedestrians. Note that lowering the overlapping threshold in NMS in order to tolerate a higher level of occlusion results in more false positives for each object. In contrast, more occluding objects are miss-detected by increasing the value of this threshold. Therefore, changing the overlap threshold for NMS heuristics would not be conducive for improving their detection performances.

---

[2]We also evaluate the detectors on synthetically generated data. Experiments showing the superior performance of our algorithm compared YOLO v2 and Faster-RCNN on this data can be found in Appendix.

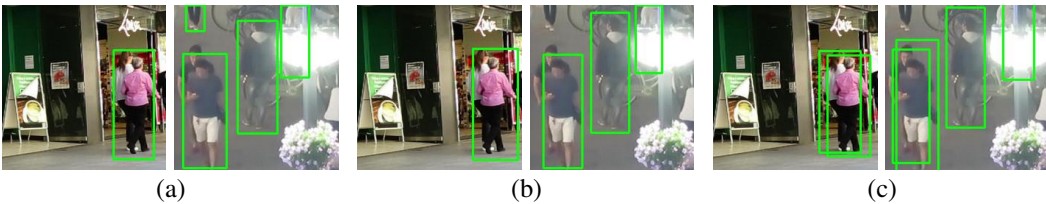

Figure 2: A comparison between the detection performance of (a) Faster-RCNN, (b) YOLO v2 and (c) our set detector on heavily overlapping pedestrians from *MOTChallenge* benchmark. Both Faster-RCNN and YOLO v2 fail to properly detect heavily occluded pedestrians due to the inevitable NMS heuristic.

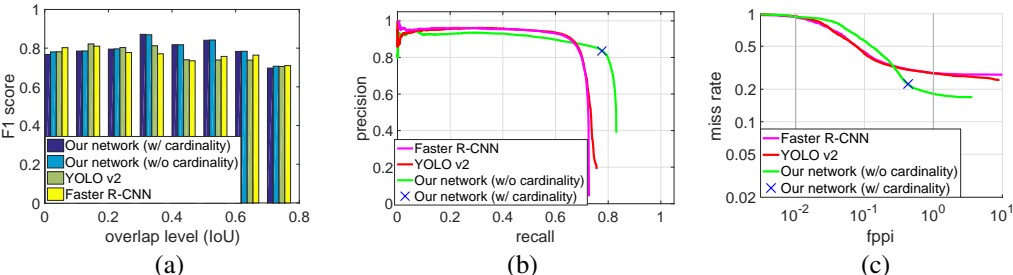

Figure 3: (a) The best F1 scores against the level of object occlusions calculated by intersection of union (IoU), (b) Precision-Recall curve, and (c) ROC (miss rate-false positive per image) curve on pedestrian detection data for the competing detectors: Faster-RCNN, YOLO v2, our network (w/o cardinality) and our network (w/ cardinality). Our final detection results are also shown as a single point in the curves.

In contrast, our set learning formulation naturally handles heavy occlusions (see Fig. 2) by outputting a set of detections with no heuristic involved. Fig. 3(a) also shows the superior performance of our formulation in detecting the objects with severe occlusion. Our method has an improvement of 5-15% in F1 score compared to YOLO v2 and Faster-RCNN for high overlap level (IoU) between $0.3 - 0.7$. The overall performance ROC and precision-recall curves for Faster-RCNN, YOLO v2 and our detector (w/o cardinality) are shown in Fig. 3(b) and (c). Note, that the single point in these curves represents our final detection result. Our approach outperforms other competing approaches with the highest F1-score and also the lowest miss rate given the same rate of false positives per image. This is significant as our set network is not yet well-developed for the detection task while YOLO v2 and Faster-RCNN are well engineered for this application.

## 4.2 CAPTCHA TEST FOR DE-SUMMING A DIGIT

We also evaluate our set formulation on a CAPTCHA test where the aim is to determine whether a user is a human or not by a complex logical test. In this test, the user is asked to decompose a *query digit* shown as an image (Fig. 4(left)) into a set of digits by clicking on a subset of numbers in a noisy image (Fig. 4(right)) such that the summation of the selected numbers is equal to the *query digit*.

In this puzzle, it is assumed there exists only one valid solution (including an empty response). We target this complex puzzle with our set learning approach. What is assumed to be available as the training data is a set of spotted locations in the *set of digits* image and no further information about the represented values of *query digit* and the *set of digits* is provided. In practice, the annotation can be acquired from the users' click when the test is successful. In our case, we generate a dataset for this test from the real handwriting MNIST dataset.

**Data generation.** The dataset is generated using the MNIST dataset of handwritten digits. The *query digit* is generated by randomly selecting one of the digits from MNIST dataset. Given a *query digit*, we create a series of random digits with different length such that there exists a subset of these digits that sums up to the *query digit*. Note that in each instance there is only one solution (including empty) to the puzzle. We place the chosen digits in a random position on a $300 \times 75$ blank image with different ro-

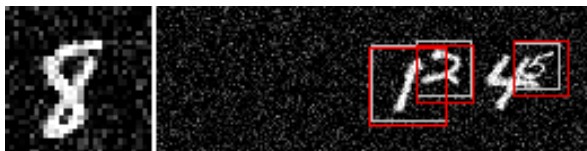

Figure 4: A *query digit* (left) and a *set of digits* (right) for the proposed CAPTCHA test. The ground truth and our predicted solutions are shown by white and red boxes respectively.

tations and sizes. To make the problem more challenging, random white noise is added to both the *query digit* and the *set of digits* images (Fig. 4). The produced dataset includes 100K problem instances for training and 10K images for evaluation, generated independently from MNIST training and test sets.

**Baseline method.** Considering the fact that only a set of locations is provided as ground truth, this problem can be seen as an analogy to the object detection problem. However, the key difference between this logical test and the object detection problem is that the objects of interest (the selected numbers) change as the *query digit* changes. For example, in Fig. 4, if the *query digit* changes to another number, *e.g.* 4, the number $\{4\}$ should be only chosen. Thus, for the same *set of digits*, now $\{1, 2, 5\}$ would be labeled as background. Since any number can be either background or foreground conditioned on the *query digit*, this problem cannot be trivially formulated as an object detection task. To prove this claim, as a baseline, we attempt to solve the CAPTCHA problem using a detector, *e.g.* the Faster-RCNN, with the same base structure as our network (ResNet-101) and trained on the exactly same data including the *query digit* and the *set of digits* images.

**Implementation details.** We use the same set formulation as in the previous experiment on object detection. Similarly, we train the same network structure (ResNet-101) using the same optimizer and hyper-parameters as described in 4.1. We do not, however, use the permutation loss $f_2(\cdot)$ since we are not interested in the permutation of the detected digits in this experiment. However, we still need to estimate the permutations iteratively using Eq. 5 to permute the ground truth for $f_1(\cdot)$.

The input to the network is both the *query digit* and the *set of digits* images and the network outputs bounding boxes corresponding to the solution set. The hyper-parameter $U$ is set to be 2, adjusted on the validation set.

**Evaluation protocol.** Localizing the numbers that sum up to the query digit is important for this task, therefore, we evaluate the performance of the network by comparing the ground truth with the predicted bounding boxes. More precisely, to represent the degree of match between the prediction and ground truth, we employ the commonly used Jaccard similarity coefficient. If $IoU_{(b1,b2)} > 0.5$ for all the numbers in the solution set, we mark the instance as correct otherwise the problem instance is counted as incorrect.

**Results.** The accuracy of our set prediction approach to solve this logical problem on the test dataset is **95.6%**. The Faster-RCNN detector failed to solve this test with an accuracy of **26.8%**. In fact, Faster-RCNN only learns to localize digits in the image and ignores the logical relationship between the objects of interest. Faster-RCNN is not capable of performing reasoning in order to generate the sensible score for a subset of objects (digits). In contrast, our set prediction formulation gives the network the ability of mimicking arithmetic implicitly by end-to-end learning the relationship between the inputs and outputs from the training data. In fact, the set network is able to generate different sets with different states and cardinality if one or both of the inputs change. We believe that this is a far more interesting outcome, as it show the potential of our formulation to tackle any other arithmetical, logical or semantic relationship problems between inputs and output without any explicit knowledge about arithmetic, logic or semantics.

## 5 CONCLUSION

In this paper, we proposed a framework for predicting sets with unknown cardinality and permutation using convolutional neural networks. In our formulation, set permutation is considered as an unobservable variable and its distribution is estimated iteratively using alternating optimization. We have shown that object detection can be elegantly formulated as a set prediction problem, where a deep network can be learned end-to-end to generate the detection outputs with no heuristic involved. We have demonstrated that the approach is able to outperform the state-of-the art object detections on real data including highly occluded objects. We have also shown the effectiveness of our set learning approach on solving a complex logical CAPTCHA test, where the aim is to de-sum a digit into its components by selecting a set of digits with an equal sum value.

The main limitation of the current framework is that the number of possible permutations exponentially grows with the maximum set size (cardinality). Therefore, applying it to large-scale problem is not straightforward and requires an accurate approximation for estimating a subset of dominant permutations. In future, we plan to overcome this limitation by learning the subset of significant permutations to target real-world large-scale problems such as multiple object tracking.

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

APPENDIX

This appendix accompanies and completes the main text. We first provide more information about the generated synthetic dataset and add more detailed results and comparison on this dataset. We also show how our approach can simultaneously detect and identify similar looking object instances across the test data due to the learning of the permutation variables. This paves the path towards end-to-end training of a network for multiple object tracking problem. Next, we include an extra baseline experiment showing the ineffectiveness of training the set network proposed in (1) for object detection problem. Finally, we present extra analysis and additional experiments on the CAPTCHA test experiment.

## 1 OBJECT DETECTION

**Synthetic data generation.** To evaluate our proposed object detector, we generate 55000 synthetic images with resolution 200x200 pixels, including 50000 images for training and 5000 for testing. For each synthetic image, we use a central crop of a randomly picked COCO 2014 image as background. The background images for our train and test sets were correspondingly taken from COCO 2014 train and validation sets in order to introduce less similarity between our train and test data. A random number of objects (from 0 to 4) is then rendered on the background. An object is a circle of radius $r$ approximated by using Bézier curves, where $r$ is randomly sampled between 25 and 50 pixels. Once the object has been created, random perturbations are applied on the control points of the curves in order to obtain a deformed ellipsoid as a result. Each object is either of red, green, yellow or blue color, with a certain variation of tone and brightness to simulate visual variations in natural images. To make the data more challenging, we allow severe overlap of the objects in the image - the intersection over union between a pair of object instances may be up to 85%. Also, random Gaussian noise is added to each image (See Figs. 5).

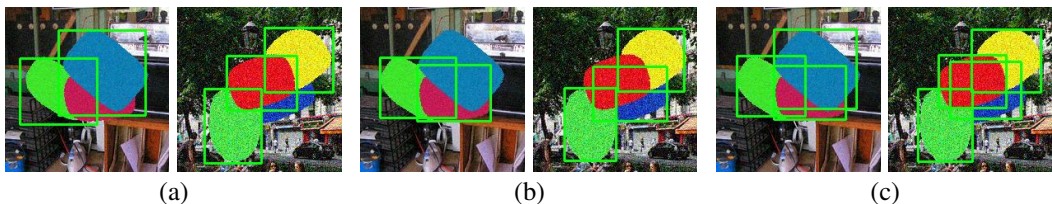

(a)           (b)           (c)

Figure 5: A comparison between the detection performance of (a) Faster-RCNN, (b) YOLO and (c) our set detector on heavily overlapping objects. Both Faster-RCNN and YOLO fail to properly detect heavily occluded objects due to the inevitable NMS heuristic.

**Detection results.** Quantitative detection results on this dataset for Faster-RCNN, YOLO v2 and our proposed detector are shown in Tab. 2.

Similar to real data scenario, our detector using the set formulation outperforms all other approaches *w.r.t.* all types of metrics. Considering the simplicity of the data and the very deep network used, *i.e.* ResNet-101, all methods work very well on localizing most of the objects with or without moderate partial occlusions. However, our set learning formulation naturally handles heavy occlusions (see Fig. 5) by outputting a set of detections with no heuristic involved.

Fig. 6(a) also shows the superior performance of our formulation in detecting the objects with severe occlusion, quantitatively (about 20% improvement in F1 score compared to YOLO v2 and faster-RCNN for overlap level (IoU) above 0.5). The ROC and precision-recall curves for Faster-RCNN, YOLO v2 and our detector (w/o cardinality) are demonstrated in Fig. 6(b) and (c). Our approach

Table 2: Detection results on the synthetic data measured by average precision, the best F1 scores (higher is better) and log-average miss rate (lower is better).

| Method | AP ↑ | F1-score↑ | MR ↓ |
|---|---|---|---|
| Faster-RCNN | 0.81 | 0.88 | 0.21 |
| YOLO | 0.86 | 0.91 | 0.17 |
| **Our Detector (w/o cardinality)** | **0.93** | 0.95 | **0.15** |
| **Our Detector** | — | **0.96** | |

outperforms other competing approaches with the highest F1-score and also the lowest miss rate given the same rate of false positives per image.

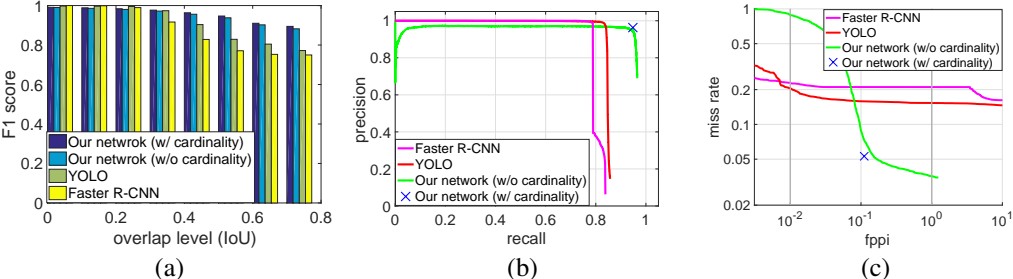

Figure 6: (a) The best F1 scores against the level of object occlusions calculated by intersection of union (IoU), (b) Precision-Recall curve, and (c) ROC (miss rate-false positive per image) curve on synthetic data for the competing detectors: Faster-RCNN, YOLO, our network (w/o cardinality) and our network (w/ cardinality). Our final detection results are also shown as a single point in the curves.

**Detection & Identification results.** In addition to the prediction of bounding boxes, our approach also provides the best ordering representation $\pi^*$ for the bounding boxes for each input instance. In this dataset, this is equivalent to finding the association between bounding boxes belonging to similarly looking instances across the different images without any knowledge about this correspondence. For example, by testing three different images in Fig 7, three sets of bounding boxes and their permutations, *e.g.* $(2, 3)$, $(2, 1, 3)$ and $(3, 4, 2, 1)$, are predicted. If the bounding boxes are re-ordered according to their permutations in a descend or ascend order, each corresponding bounding boxes in each set will be associated to the similarly looking objects in each image.

Therefore, we can simultaneously detect and identify the similarly looking instances for different test examples, *e.g.* for each frame of a video sequence.

Note that this problem cannot be formulated as a classification problem as we do not assume a fixed labelling representation for set instances. In fact, the number of identifiable unique instances only depends on the maximum number of set elements outputted from our network.

For evaluation purposes only, we use the associations between similarly looking objects

Figure 7: The performance of our approach in detecting and also identifying the object instances using permutations. The similar colours' boxes identify the same objects. Here we used color coding to represent the same ID.

which are available from our simulated data. For example, all bounding boxes belonging to the red object within the dataset with all its visual variations can be assumed to be the same instance. Therefore, we can evaluate our identification performance using this information. To assess the results, we use the permutation accuracy. We achieve $81.1\%$ accuracy in identifying the similarly looking objects. Fig. 7 shows the detection and identification results for three different images.

This joint detection and identification of the objects can be a fundamental step toward end-to-end training of a network for the multiple object tracking problem. However, the current formulation cannot handle large-scale object tracking problems due to the exponential number of permutations. Making the problem tractable for many targets and incorporating motions is left for future work.

**An additional baseline experiment.** We performed an experiment by training the set network proposed in (1) with the same base network structure (ResNet-101) on the synthetic data reported above. In contrast to our networks' losses (Fig. 8 (a)), the training and validation losses for the set network (Fig. 8 (b)) cannot decreases beyond a very high value and the generated boxes are simply an average of the positions and sizes (Fig. 9), proving the ineffectiveness of the approach to learn a reasonable model for this task. As mentioned before, the reason for this failure is that the permutations are not considered in the model. Therefore, when the network is trained with orderless outputs such as bounding boxes, the model is confused. Thus, its objective function does not converge when forcing it to learn from such data.

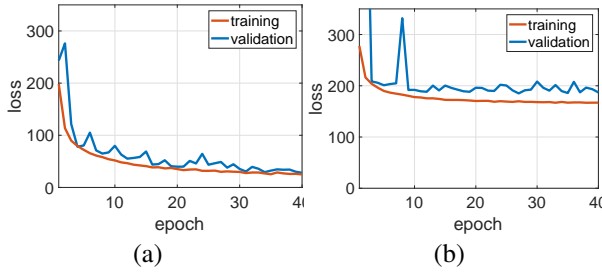

Figure 8: Training and validation losses of (a) our Deep Perm-Set Network and (b) the set network proposed in (1), for object detection task.

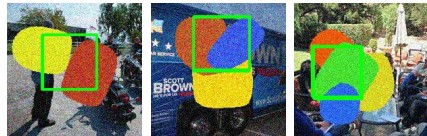

Figure 9: The predicted bounding boxes from the set network proposed in (1), when trained for the object detection task. In all images, all the bounding boxes are concentrated in the same location with the exactly same size, which are simply an average of all the positions and sizes of the objects in the scene.

## 2 CAPTCHA TEST FOR DE-SUMMING A DIGIT

For solving the CAPTCHA test using Faster-RCNN, we investigated if changing the detection threshold can significantly improve the accuracy of the test. As shown in Fig. 10, this threshold has a slight effect in the accuracy of test, increasing it from **26.8%** for a default detection threshold 0.5 to **31.05%** for the best threshold.

We also explored if an additional network, trained to recognize the digits, can be used in combination with Faster-RCNN in order to improve the accuracy of the test. To this end, we used another ResNet-101 network and trained in MNIST dataset to recognise the digits. Note that we assumed that the annotations for our dataset are only the bounding boxes and no information about the *query digit* and the *set of digits* are provided. Therefore, this new classifier network could not be trained on our dataset.

In order to solve the test with an additional digit classifier, both the *query digit* and the detected numbers from Faster-RCNN are fed to the classifier network to recognise their values. Then, a subset of the detected digits (including an empty set) is chosen to solve the test. It is obvious that the test accuracy is affected by the performance of both detection and classifier networks. For this experiment, the test accuracy is increased considerably from **31.05%** to

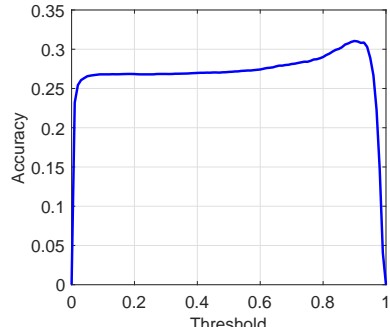

Figure 10: A plot, representing the accuracy of solving the CAPTCHA test using Faster-RCNN against the detection threshold.

**59.28%**. However, it is still significantly beyond our reported results for this test (**95.2%**). Figure 11 shows more results from our approach to solve this test. The output of Faster-RCNN for the examples are also shown in Figure 12.

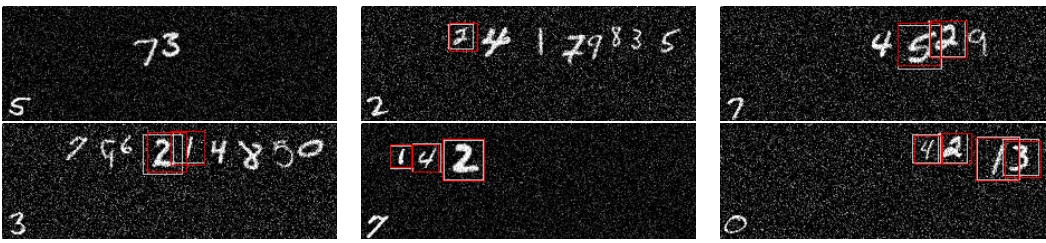

Figure 11: Further examples showing a perfect prediction of the solution using our Deep Perm-Set Network. Each image contains a *query digit* (bottom left corner) and a *set of digits*. The ground truth and our predicted solutions are shown by white and red boxes respectively. Note that *zero* represents number 10 for our experiment.

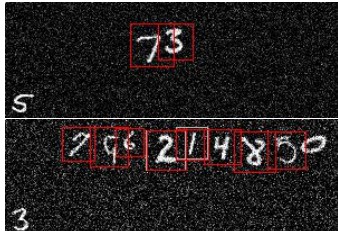 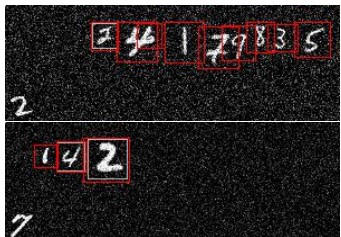 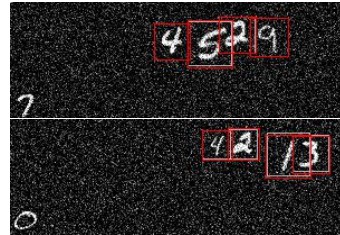

Figure 12: Further examples showing the solution of Faster-RCNN for the test. Each image contains a *query digit* (bottom left corner) and a *set of digits*. The ground truth and the predicted solutions for Faster-RCNN are shown by white and red boxes respectively. Faster-RCNN simply learns to detect almost all the digits while ignoring the logical relationship between the *query digit* and the *set of digits*.

REFERENCES

[1] S. Hamid Rezatofighi, Anton Milan, Qinfeng Shi, Anthony Dick, and Ian Reid. Joint learning of set cardinality and state distribution. In *AAAI*, 2018.

