# OpenReview forum: "Deep Perm-Set Net: Learn to predict sets with unknown permutation and cardinality using deep neural networks"
_ICLR.cc/2019/Conference_

### Official Review · AnonReviewer2 · 2018-11-02
**Lacks clarity of methodology.**

**Rating:** 3
**Confidence:** 4

**Review:**

This paper looks to predict "unstructured" set output data. It extends Rezatofighi et al 2018 by modeling a latent permutation.

Unfortunately, there is a bit of an identity crisis happening in this paper. There are several choices that do not follow based on the data the paper considers.
1) The paper claims to want to predict unordered sets, yet the model is clearly indicating a dependence in the order of the outputs and the input p_m(\pi | x_i, w) (1); this feels like a very odd choice to me. The outputs are either unordered sets, where you would have a permutation invariant (or exchangeable) likelihood, or they are ordered sequence where the order of the outputs does matter, as some are more likely than others.
2) The paper still makes very odd choices even if one ignores the above and wants to model some orderings as more likely than others. The way the permutation, or the order of the data, accounts in the likelihood (2) does not make sense. Conditioned on the permutation of the set, the points are exchangeable. Let's just consider a 2 element "set" at the moment Y = (y_1, y_2). Order matters, so either this is being observed as pi=(1, 2) or pi=(2, 1), both of which depend on the input x. However, the likelihood of the points does not actually depend on the order in any traditional sense of the word. we have:
p_\pi((1, 2) | x, w) p_y(y_1 |  x, w, (1, 2)) p_y(y_2 |  x, w, (1, 2)) + p_\pi((2, 1) | x, w) p_y(y_1 |  x, w, (2, 1)) p_y(y_2 |  x, w, (2, 1))
*Note that in here (as in eq. 2) the output distribution p_y does not know what the index is of what it is outputting, since it is iid.* So what does this mean? It means that the order (permutation) can only affect the distribution in an iid (exchangeable, order invariant) way. Essentially the paper has just written a mixture model for the output points where there are as many components as permutations. I don't think this makes much sense, and if it was an intentional choice, the paper did a poor job of indicating it.
3) Supposing even still that one does want a mixture model with as many components as permutations, there are still some issues. It is very unclear how the dependence on \pi drops out when getting a MAP estimate of outputs in section 3.3. This needs to be justified.

There are some stylistic shortcomings as well. For example, the related works paper would read better if it wasn't one long block (i.e. break it into several paragraphs). Also, the paper claims that it will use a super script m to denote a known cardinality, yet omits \mathcal{Y}_i^{m_i} in the training set of the first sentence in 3.1. But these and other points are minor.

The paper should not be published until it can resolve or make sense of the methodological discrepancies between what it says it looks to do and what it actually does as described in points 1), 2), and 3) above.

---

> ### Public Comment · ~Maxwell_B_Greason1 · 2018-11-07
> **The purpose of the permutation component**
>
> I am not affiliated with the authors of this paper, but I spent quite some time puzzling over it myself. The question of "if it's for unordered sets, why on Earth does it need to bother with predicting the permutation in the first place?" puzzled me as well, but I think I eventually figured it out.
>
> It's for aligning the output with the training labels. That's why the permutation element only matters during training and is ignored at inference! If you want to properly score two sets against each other, you have to decide which element should be compared with which; if you're predicting, say, sets of numbers and you predict {10, -3} when the ground-truth labels are {-4, 9}, then how you score the model will depend greatly on how you align each element of the sets. It matters quite a bit for the loss - and the resulting gradient - if you view the 10 as a pathetic failure to predict that the set should contain a -4, or if you instead treat it as a nearly-accurate prediction of the set containing a 9.
>
> So in order to fairly score the set output versus the ground truth set, you essentially have to examine every possible way of matching up output elements to input elements in order to select the one which gives the closest match. This would be staggeringly computationally expensive, so they settle for just trying to guess what permutation will line up with the training data best, and then directly scoring that permutation against the ground-truth labels and hoping that comes close enough.
>
> In the DeepSets paper (https://arxiv.org/abs/1703.06114), this is why it was limited to only producing sets of the same cardinality and permutation as the input set; because that way, you can directly track the attribution and determine which label properly belongs to which input element, and thereby score the corresponding outputs on how close they got to that label.
>
> (In other words: the permutation matters only for credit assignment purposes.)

---

> > ### Author Response · Authors · 2018-11-12
> > **Thank you**
> >
> > We appreciate your clarification.

---

> ### Author Response · Authors · 2018-11-12
> **Clarifying the formulation**
>
> We thank Maxwell for some clarification. We believe AnonReviewer2 misunderstood some of the concepts and we will try to clarify them here and update the paper accordingly.
>
> - predicting unordered sets
> The assumption is what is available as GT is a set. This means we cannot infer any specific ordering from GT. The proposed framework is very flexible as we don’t need to enforce the problem to be necessarily orderless  (although it can be). The reason we would like to learn  p_m(\pi | x_i, w) is to infer the nature of the problem. However, excluding the main experiment in supplementary material, we did enforce the problem to be orderless by removing O2 and the permutation loss. This is equivalent to assume p_m(\pi | x_i, w) is uniform (order does not matter) in Eq.2 and you can see O2 and its loss will be eliminated from Eq. 5 and 6. However, we still require to solve Eq. 5 to find the best permutation based on f1 only, which is equivalent to use Hungarian to solve the assignments.
>
> We also disagree with R3 that the problem is either unordered sets or there exist only one order to be correct. There can exist multiple orders to be true, but not all. This can be inferred by learning p_m(\pi | x_i, w) from samples derived during training by Eq. 5.
>
> - permutation in the likelihood (2) does not make sense:
> In addition to what is explained by Maxwell, I add this clarification:
> p_y(y_1 | x, w, (1, 2)) means the first output is assigned to the first ground truth, while p_y(y_1 | x, w, (2, 1)) mean the first output is assigned to the second ground truth. These two scenarios are acctally generate very different gradient.  The same argument can be extended to p_y(y_2 | x, w, (1, 2)) and p_y(y_2 | x, w, (2, 1)).
>
> - the dependence on \pi drops out when getting a MAP estimate of outputs:
> The permutation takes into the account when there is loss and a GT to compare as GT  annotations are permutated to be assigned to the outputs. During inference, we don’t have loss and GT. We just have the predicted outputs, e.g. cardinality, states and premutation and the order which we want to show the states will not change the value of the states.
>
>
> We hope to have clarified all the technical misunderstandings. We would like to point the reviewer again to our impressive results in the detection problem and ask him/her to reconsider his/her rating if the technical concerns are now clear.

---

> > ### Comment · AnonReviewer2 · 2018-11-30
> > **Poor Clarity**
> >
> > Misunderstandings are due to poor clarity in the text as written.
> >
> > "... (we) assume p_m(\pi | x_i, w) is uniform," this is very unclear in the text.
> >
> > I also appreciate that one may want to make use of a permutation to match outputs to fixed predictions. However, I again invite the authors to stare at what they have written (eq. 6): p(y_\sigma | x_i, w, π). Note that the model does not have access (it is not conditioned on) to the indices \sigma, it is completely exchangeable. As I explained, this boils down to a mixture model. Now, what it sounds like the authors wanted to write (to match things up) is:  p(y_\sigma | x_i, w, π, \sigma). It's a small change that makes a big difference, and illustrates the sort of clarity issues that abound the paper.
> >
> > (There is also something to be said that a sequential auto-regressive methods can already keep track of what has been seen, and doesn't require any matching of permutations. This was my point about having a truly orderless model, that does not need order matching, or just having a sequential method.)
> >
> > I'm glad that the authors are seeing good performance and seem to have an effective method for matching outputs to fixed predictions, however the quality of the paper is too poor for publication.

---

### Official Review · AnonReviewer1 · 2018-11-02
**Interesting results on object detection for overlapping objects and CAPTCHA toy problem, but key details seem unclear**

**Rating:** 3
**Confidence:** 3

**Review:**

— Summary
The method extends [21], which proposes an unordered set prediction model for multi-class classification. For that problem, [21] can assume logistic outputs for all distinct classes. This work extends set prediction to the object detection task, where box identity is not distinct — this is handled by an additional model output that reasons about the most likely object permutations. The permutation predictions are used during training, but are not needed at inference time — as shown in Fig1 and Eq 7. Results are on detection of overlapping objects and a CAPTCHA toy summation example.

— Clarity
The exposition is not particularly clear in several places:
 - U^m in Eq 1 is undefined and un-discussed. What probability term does it correspond to? It is supposed to make probabilities of different cardinalities comparable, but the exact mechanism is unclear.
 - The term p(w) disappears on the left hand side of Eq 2.
 - Notation in Sec. 3.2 is very cumbersome, making it hard to follow. Furthermore, I found the description ambiguous, preventing me from understanding how exactly the permutation head output is used in Eq 5. Specifically, there is some confusion about estimation of w~, which seems based on frequency estimation from past SGD iterations (Eq 3). If so, why does term f2 in Eq 5 contain the permutation head output O2 and how do the two relate?
 - The network architecture is never described, especially the transition from Conv to Dense and the layer sizes, making the work hard to reproduce. The dimensions of the convolutional feature map matter (probably need to be kept tractable).

— Significance
Key aspects of the model are not particularly clear, specifically about how the permutation prediction ( the key novelty here) is used to benefit training.
— Term f2 in Eq5 uses w~ estimates, which appeared to be based on statistics from past SGD runs, yet also depends on the output of the permutation head O2. Am I misinterpreting the method?
— In the paragraph right after Eq5, it’s claimed that “Empirically, in our applications, we found out that estimation of the permutations from just f1 [in Eq5] is sufficient to train properly … by using the Hungarian algorithm”. So then f2 term is not even used in. Eq5? If so, what is the significance of the permutation head other than adding an auxiliary loss?

Furthermore, there are no experimental results demonstrating the effect of the permutation head and the design choices above — if we could get by with only using the Hungarian algorithm, why bother classifying an exponential number of permutations? Do they help when added as an auxiliary loss?

While the failure of NMS to detect overlapping objects is expected, the experiments showing that perm-set prediction handles them well is interesting and promising. Solving the general case with larger images and many instances would increase the impact significantly — and likely require a combination of perm-set prediction and image tiling, although this is just a hypothesis. The Captcha toy example also shows some interesting behavior emerging — without digit-specific annotations (otherwise it would be multi-class classification setup from [21]), the model can handle the majority of summations correctly.

— Experimental results
The results are interesting proofs-of-concept but a few more experiments/answers would be helpful:
- It still appears that PR curve in the high-precision regime (fig 3b) has lower precision than FRCNN/YOLO. Any idea as to why?
- Ablation results on the effect of the permutation predictions vs Hungarian algorithm, etc would be helpful, as discussed above.
- How sensitive is the method to seeing a certain cardinality? What if it never sees 3 pedestrians in an image, but only 1,2,4 will it fail to predict 3? Or alternatively, if we train a model that can handle up to 5-6 entities with examples than have <=4? What is the right way of data augmentation for this model (was there any and should there be?)
- Given that values for U differ across applications, how sensitive is the output / how much sweeping did you have to do?

-- Related work
To the best of my knowledge it's representative. It would help to cite more recent work that decreases detector dependence on NMS. For example, "Learning Non-Maximum Suppression", Hosang, Benenson, Schiele, CVPR 2017 or "Relation Networks for Object Detection", by Hu et al, CVPR 2018 and references therein.

---

> ### Author Response · Authors · 2018-11-12
> **Clarifying all key details**
>
>  U^m in Eq 1:
> This term is clearly defined at the beginning of section 3, first paragraph, line 6 “U is the unit of hyper-volume in ...” . U^m is simply U powered by the cardinality variable.
> In section 3.3, in the paragraph after Eq.7 line 2, we explain the mechanism to obtain U. For each experiment, we also report the tuned value for U.
>
> - The term p(w) in Eq 2:
> Note Eq. 2 calculates the posterior, i.e., p(w|D) and according to the Bayes theorem, it is p(w|D) \propto p(D|w)p(w) which is simply Eq. 1.
>
> - Eq 5 confusion :
> To explain Eq.5 and Eq. 6, the training works as follows:
> At each iteration k-1, the network predicts the outputs, i.e., O1, O2 and \alpha. We first solve a discrete optimization to find the permutation (matching) between the predictions at k-1 and the ground truth (GT). Then, we use this permutation to order GT and back-propagate the losses to update w at iteration k. Please note that cardinality loss does not depend on this permutation variable.
>
> - The network architecture :
> Note that our described methodology can be applied to any network architecture. In ALL our experiments, we use Res-net 101 (mentioned several times on page 6 and 8).
> We only need to define the number of outputs and use the set loss defined in Eq. 5 and 6. For example, for the set size with maximum cardinality 4, we need 5 outputs ( \alpha) for cardinality m = {0, 1,...4}. If the state of each set is 5 for the detection experiment, we need 4*5=20 outputs for the state  loss (O1). For the permutation (O2), we need 4!=24 outputs.
> For each output we have a loss defined for each experiment in the text.
>
> - the permutation to benefit training:
> We refer the reviewer to the experiment we have already included in Appendix titled “An additional baseline experiment”, which unfortunately we could not include in the main manuscript due to space constraints.
>  We use a baseline model with no permutation, which is exactly same as [21], to train the network for the detection task. The results show the model is not able to learn this task, hence highlighting the need for permutation prediction for a complete set prediction network.
> Even if we remove the permutation head, O2, from our model, we still need to calculate the permutation using Eq. 5 and use it for backpropagation in Eq. 6. However the model in [21] completely ignore the permutation in its formulation. Therefore, it cannot learn the detection task.
>
> - Term f2 in Eq5 uses w~ estimates:
> Your interpretation is indeed correct. Given the predictions of O1 and O2 using statistics from past SGD runs, we want to find the best permutation. There are indeed m! way to assign GT set elements to the predictions. We solve this optimization to find the best one.
>
> - the significance of the permutation:
> Even if we don’t use f2 for the estimation of the best permutation, we can use \pi* as ground truth for updating its loss in Eq. 6.
>
> - classifying permutations:
> Classifying the permutations provides the extra information about the structure of the problem, e.g. there exist a single order which matters or it can be several different orders or the problem is orderless. We simply do not ignore the permutations from Hungarian by allowing the network to learn them. We refer you to the experiment we have already included in Appendix titled “Detection & Identification results”, where we used the predicted permutations to identify the bounding boxes for similar looking objects across different test images
>
> -  larger images and many instances:
> We agree. We leave this as future work, as it would require an engineering effort that departs from the main purpose of our paper, which is to show theoretically how to construct a network that can work with sets instead of tensors.
>
> - sensitivity to seeing a certain cardinality:
> During the training, we do the data augmentation (by cropping, flipping etc) to ensure the network sees enough sample for each cardinality. We will include this detail in the text.
>
> - Related work
> We are happy to include these references. But these works are orthogonal to the main subject of our paper. In our paper the goal is to introduce a framework to output a set using neural networks and we used the detection task as one of the set prediction examples.
> We would like to add few comment about these two works:
> These approaches try to learn a pairwise relationship between the boxes outputted using the conventional proposal based detection approaches. First a) they need to introduce extra pairwise network or heavier computation to learn these pairwise relationship b) they assume the relationship is pairwise between bounding boxes. Out framework is a single stage approach which uses a conventional convnet backbones with no extra computation. Since it is end-to-end prediction of boxes, we don’t enforce any pairwise or higher order relationships between the outputs. We all rely on the layer of neural nets to capture high level relationship between outputs before predicting them.

---

> > ### Comment · AnonReviewer1 · 2018-11-26
> > **keeping my rating, despite helpful clarifications**
> >
> > Thank you for clarifying details I asked about. Specifically, U^m, p(w|D) in Eq 2 and the network architecture. On the latter please note that it's not obvious whether you add any extra FC layers between ResNet and your outputs. ResNet typically does not have these Dense layers (excluding the softmax) yet they are shown in Fig1. What are their sizes?
> >
> > I find the paper still too scattered, trying to solve diverse problems with a hammer without properly motivating / analyzing key details of this hammer. So I keep my rating.
> >
> > Significance of the permutation head (the main technical contribution, really). Ablation would illuminate the impact of the modeling choices made in this paper:
> > -- You point to a comparison to [21] in the appendix, however [21] seems completely permutation-unaware so it understandably fails. I was expecting a comparison with  hungarian matching for the detection case -- what happens if you completely omit f2 in Eq5.  You suggest this yourself in your feedback -- why is there no experimental ablation of this key detail? Additionally, what happens if vanilla Hungarian matching is done using simply a box overlap metric in Eq 5 -- this is essentially what anchor boxes do, and can be a standard baseline.
> > -- If we have Hungarian matching working, we can completely remove the permutation head from the detection network and have something that trains potentially fine? How well does this work?
> > -- What is exactly the effect of having the weight w~tilde from Eq 3 estimated from past SGD iteration frequency. What if I omit it or what if I use the posterior of head O2?
> >
> > It is true that doing set prediction goes beyond the detection example -- as illustrated in the two toy examples -- there are cases indeed where the permutation head can be directly useful as output, but those do remain toy examples (one of them is entirely in the appendix so it carries limited weight).

---

### Official Review · AnonReviewer3 · 2018-11-05
**novel, potentially significant & with limitation on large-scale problem**

**Rating:** 7
**Confidence:** 3

**Review:**


The paper is really interesting. Set prediction problem has lots of applications in AI applications and the problem has not been conquered by deep networks.

The paper proposes a formulation to learn the distribution over unobservable permutation variables based on deep networks and uses a MAP  estimator for inference.  It has object detection applications. The results show that it can outperform YOLOv2 and Faster R-CNN in a small pedestrian detection dataset which contains heavy occlusions.

The limitation is clearly stated in the last part of the paper that the number of possible permutations exponentially grows with the maximum set size (cardinality).

In the author response period, I would like the author give more details about the pedestrian detection experiments, such as how many dense layers are used after ResNet-101, what are the training and inference time, is it possible to report results on PASCAL VOC (only the person class).

The method is exciting for object detection funs.  I would like to encourage the authors to release the code and let the whole object detection community overcome the limitation in the paper.

---

> ### Author Response · Authors · 2018-11-12
> **Network details and inference time**
>
> We appreciate AnonReviewer3’s recognition of our work.
>
> - Network details
> We only replace the last fc layer of ResNet-101 with a new fc layer mapping to 49 (5+20+24 = 49) outputs for calculating cardinality, states and permutation (the choice of these numbers explained in our response to R2).
>
> - inference time
> We also performed extra experiment on accuracy and inference time between different detectors (on the same machine and GPU) reported here:
>
> Faster R-CNN: AP=0.68, Inference time=101 ms
> YOLO v2: AP=0.68, Inference time=12.3 ms
> YOLO v3: AP=0.70, Inference time=18.2 ms
> Our network: AP=0.75, Inference time=15.1 ms
>
> - test on PASCAL VOC
> We observed PASCAL VOC dataset include many images with more than 4 persons. Considering the images include up to 4 persons only, we might not have enough training data to train ResNet-101 network.

---

### Meta-Review · Area_Chair1 · 2018-12-16
**Area chair recommendation**

**Confidence:** 5
**Recommendation:** Reject

**Metareview:**

Strengths:
The method extends [21], which proposes an unordered set prediction model for multi-class classification.
The submission proposes a formulation to learn the distribution over unobservable permutation variables based on deep networks and uses a MAP  estimator for inference.
While the failure of NMS to detect overlapping objects is expected, the experiments showing that perm-set prediction handles them well is interesting and promising.

Weaknesses:

Reviewer 1: "I find the paper still too scattered, trying to solve diverse problems with a hammer without properly motivating / analyzing key details of this hammer. So I keep my rating."
Reviewer 2: "I'm glad that the authors are seeing good performance and seem to have an effective method for matching outputs to fixed predictions, however the quality of the paper is too poor for publication."

Points of contention:

 Although there was one reviewer who gave a high rating, they were not responsive in the rebuttal phase.  The other two reviewers took into account the author responses, and a contributed comment by an unaffiliated reviewer, and both concluded that the paper still had serious issues.  The main issues were: lack of clear methodology and poor clarity (AnonReviewer2), and poor organization and lack of motivation for modeling choices (AnonReviewer1).